# Phylogeography and Antioxidant Activity of Proso Millet (*Panicum miliaceum* L.)

**DOI:** 10.3390/plants10102112

**Published:** 2021-10-05

**Authors:** Xiao-Han Wang, Myung-Chul Lee, Yu-Mi Choi, Seong-Hoon Kim, Seahee Han, Kebede Taye Desta, Hye-Myeong Yoon, Yoon-Jung Lee, Mi-Ae Oh, Jung-Yoon Yi, Myoung-Jae Shin

**Affiliations:** 1National Agrobiodiversity Center, National Institute of Agricultural Sciences, Rural Development Administration, Jeonju 54874, Korea; wangxiaohan0530@gmail.com (X.-H.W.); mcleekor@korea.kr (M.-C.L.); ymchoi@rda.go.kr (Y.-M.C.); shkim0819@korea.kr (S.-H.K.); kebetila@gmail.com (K.T.D.); mmihm@korea.kr (H.-M.Y.); yoon112@korea.kr (Y.-J.L.); miae9550@korea.kr (M.-A.O.); naaeskr@korea.kr (J.-Y.Y.); 2Honam National Institute of Biological Resources, Mokpo 58762, Korea; seahee113@hnibr.re.kr; 3Department of Applied Chemistry, Adama Science and Technology University, Adama 1888, Ethiopia

**Keywords:** gene migration, population structure, genetic diversity, association analysis, SSR marker, total phenolic content

## Abstract

Proso millet (*Panicum miliaceum* L.) or broomcorn millet is among the most important food crops to be domesticated by humans; it is widely distributed in America, Europe, and Asia. In this study, we genotyped 578 accessions of *P*. *miliaceum* using 37 single-sequence repeat (SSR) markers, to study the genetic diversity and population structure of each accession. We also investigated total phenolic content (TPC) and superoxide dismutase (SOD) activity and performed association analysis using SSR markers. The results showed that genetic diversity and genetic distance were related to geographic location and the fixation index (Fst). Population structure analysis divided the population into three subpopulations. Based on 3 subpopulations, the population is divided into six clusters in consideration of geographical distribution characteristics and agronomic traits. Based on the genetic diversity, population structure, pairwise Fst, and gene flow analyses, we described the topological structure of the six proso millet subpopulations, and the geographic distribution and migration of each cluster. Comparison of the published cluster (cluster 1) with unique germplasms in Japan and South Korea suggested Turkey as a possible secondary center of origin and domestication (cluster 3) for the cluster. We also discovered a cluster domesticated in Nepal (cluster 6) that is adapted to high-latitude and high-altitude cultivation conditions. Differences in phenotypic characteristics, such as TPC, were observed between the clusters. The association analysis showed that TPC was associated with SSR-31, which explained 7.1% of the total variance, respectively. The development of markers associated with TPC and SOD will provide breeders with new tools to improve the quality of proso millet through marker-assisted selection.

## 1. Introduction

Proso millet (*Panicum miliaceum* L.) is an annual monocotyledonous grass crop. Archaeological evidence indicates that this crop was first domesticated in northern China about 10,000 years ago [1]. Today, proso millet is widely distributed in the Americas, Europe, and Asia, and is still among the most important food crops worldwide [2]. Proso millet has a short growth cycle and low water requirement; rotation with proso millet can maintain moisture in deep soil layers, control winter weeds, and reduce the occurrence of pests and diseases, making it an ideal rotation crop for winter wheat [3]. When other crops fail to harvest or planting is delayed due to adverse weather, proso millet can be planted as an intercropping crop to reduce economic losses [4]. Proso millet is also widely used in the bird, pet feed, snack food, and wine-making industries [5]. At present, the demand for proso millet is highest for bird feed production [6].

Many previous studies have examined the genetic diversity of proso millet, including in China (using 88 accession core collections selected from 8,515 materials based on 67 proso millet-specific single-sequence repeat (SSR) markers) [5], and Canada and the USA (using 12 accessions based on amplified fragment length polymorphisms [AFLPs]) [7], as well as in six countries using 50 accessions based on 25 SSR markers [8], and 25 countries using 90 accessions based on 100 SSR markers [6]. However, these studies focused on explaining the genetic diversity and clades of local proso millet populations.

The evolutionary origin of proso millet has always been controversial, with domestication centers being proposed in multiple regions of China and Eastern Europe [9], and in a single center in China [10]. Accumulating evidence from archaeological, diversity and phylogenetic studies, among others, suggests that that proso millet originated from carbonized grains about 10,000 years ago; these grains were unearthed at the Cishan site in China [1]. The Dadiwan site in the Loess Plateau, and the Xinglonggou site in Inner Mongolia, have also yielded carbonized proso millet particles believed to be about 8000 years old [11,12]. Another study suggested that the European proso millet first appeared about 7000 years ago [13]. This was revised to 3600 years ago based on direct measurement of the crop remains [14]. However, it is difficult to determine the place of origin of proso millet based on the distribution of its wild ancestors, because this species is easily back-mutated from domesticated crops to weeds [13]. A study using SSR markers to genotype domesticated proso millet in China concluded that genetic diversity in China was highest on the Loess Plateau [10]. One study used 98 accessions from Eurasia and 16 SSR markers to explore the possibility that Eastern Europe is one of several sites of origin of domesticated proso millet [9]. However, due to the limited availability of accessions, comprehensive analysis of genetic diversity is difficult and observer bias can affect diversity analysis. 

Several studies have reported benefits of proso millet polyphenols, such as anti-inflammatory effects [15], anti-proliferative effects in colorectal cancer [16], liver protection due to syringic acid [17], free radical scavenging by ferulic acid, and antioxidant activity [18]. Proso millet varieties differ in terms of free syringic acid, ferulic acid, chlorogenic acid, caffeic acid, and *p*-coumaric acid content [19]. Total phenolic content (TPC) mainly depends on the variety, rather than type or color, of proso millet, although TPC and antioxidant capacity are also significantly affected by climatic and environmental conditions [20]. 

Breeders use genetic markers to assess the phenotypes of target traits in the early stages of the growth cycle; this approach greatly shortens the research cycle and reduces the workload associated with crop breeding. Molecular markers are essential for improving traits that cannot be directly measured. The development of molecular markers for phenols has allowed rapid estimation of the types and quantity of phenolic compounds in individual plants. however, no studies have developed molecular markers of TPC and antioxidants in proso millet, although such markers have been developed for other plant taxa. For example, molecular markers were developed to identify phenolic compounds in wild and cultivated barley [21], and a genome-wide association study identified 11 quantitative traits related to nucleosides in snap bean [22]. Genome-wide association studies have revealed that apple polyphenols are controlled by 4-*O*-caffeoylquinic acid and procyanidins B1, B2, and C1, and demonstrated the applicability of these markers to marker-assisted breeding. Although polyphenols are important components of the human diet, breeders have not widely regarded them as breeding targets; however, polyphenolic enhancement of nutritional properties may become a future breeding trend. With the improvement of the cultivation environment and the impact of cash crops, locally endemic varieties of proso millet are rapidly disappearing, which greatly affects the diversity of proso millet populations. Therefore, as genetic important resources, proso millet and other small grain crops have been a focus of research.

The objective of this study was to explore the phylogeography of proso millet, compare the TPC and antioxidant properties of 578 proso millet accessions collected in 17 countries of origin of the world, and identify SSR markers associated with these traits for use in marker-assisted breeding to enhance nutritional properties. This study has developed 37 EST-SSR markers suitable for proso millet, which should improve the accuracy of genetic diversity analyses of this species and improve our knowledge of the migration events and degree of differentiation among its evolutionary origin centers.

## 2. Results

### 2.1. Genetic Diversity Analysis

In a preliminary experiment, proso millet samples were amplified; there were 481 expressed sequence tag (EST) SSR markers, 37 EST-SSR markers that can be successfully amplified in the DNA of all individuals and have polymorphisms were screened out (Appendix A). Among 578 proso millet accessions collected in 17 countries of origin, 37 pairs of SSR primers were used to amplify 151 alleles (Appendix A).

Genetic diversity analysis showed that, among the 578 germplasms, the number of alleles (Na) per locus ranged from 2 to 7, with an average of 4.0811. The number of amplified genotypes (Ng) ranged from 2 to 17, with an average of 4.7838. Shannon’s information index (I) ranged from 0.0803 to 1.436, with an average of 0.387. Observed heterozygosity (Ho) ranged from 0 to 0.0657, with an average of 0.0032, indicating gene flow among individuals and genotypes. The genetic diversity value (H) ranged from 0.0274 (SSR-143) to 0.7331 (SSR-365), with an average of 0.19. The fixation index (Fst) is used to measure the proso millet population genetic differentiation. Among the 37 SSR markers, each marker provided a different ability to distinguish genetic differentiation, ranging from 0.0452 (SSR-458) to 0.6783 (SSR-128), with an average of 0.4545. The polymorphic information content (PIC) value of the SSRs ranged from 0.0273 (SSR-143) to 0.6884 (SSR-365), with an average of 0.1735. The average major allele frequency (MAF) was 0.8740, with a range of 0.3789 (SSR-365) to 0.9862 (SSR-143). Three SSRs showed high PIC values, SSR-203 (0.5086), SSR-232 (0.5944), and SSR-365 (0.6884), i.e., values exceeding the critical value of 0.5. The detailed parameters are listed in Table 1 and the genetic diversity analysis results for each place of origin are listed in Table 2. Genetic resources from Ukraine showed the highest diversity index (0.247 ± 0.247), followed by Russia (0.215 ± 0.183) and South Korea (0.156 ± 0.192).

Pairwise Fst is used to evaluate the degree of genetic differentiation among 17 countries of origin (Table 3). Pairwise comparisons between accessions from different origins showed that the Fst values ranged from 0.027 between Czechoslovakia (CSK) and Kazakhstan (KAZ) to 0.2699 between Turkey (TUR) and France (FRA). The lowest differentiation was observed between the population composed of germplasm originating from Korea and 17 other populations, 3 and 14 of which showed low (Fst = 0.05) and moderate (Fst = 0.15) differentiation. The degree of differentiation among the Russian (RUS), Ukrainian (UKR), and Chinese (CHN) groups was lower than among other groups. A comparison of populations native to China with other populations showed that 14 populations had a moderate degree of differentiation, whereas 3 had a high degree of differentiation (Fst = 0.15–0.25): Bolivia (BOL), FRA, and Tajikistan (TJK). The TUR population showed the highest degree of differentiation from FRA (Fst = 0.2699), followed by the Iranian (IRN), North Korean (PRK), and South Korean (KOR), and CHN (Fst = 0.1) populations. TUR, TJK, BOL, CSK, and Azerbaijan (AZE) showed the greatest within-country differentiation among populations. The results of the global Fst for all populations show that UZB, IND, RUS, IRN, and CSK are all low genetically differentiated. The populations of KAZ, FRA, PRK, MNG, and TJK show a moderate degree of genetic differentiation. The populations of CHN, BOL, THA, KOR, NPL, TUR, AZE and UKR show high genetic differentiation.

### 2.2. Population Structure and Phylogenetic Analysis

We performed population structural analysis using the Structure Harvester program [23], with the number of subpopulations (K) ranging from 2 to 20. The results showed that ΔK reached a maximum (16.65391) at K = 3, indicating that this was the most suitable K (Figure 1a), followed by 10 (ΔK = 11.57047), 8 (5.402586), and 6 (3.059784). The 578 accessions from various regions were therefore classified into three subgroups (Figure 1b). Individuals of the three subgroups were widely distributed in Eurasia (Figure 1c), of which subpopulations B (orange) and C (azure) were distributed in Eurasia, and subpopulations A (dark blue) in Asia.

Phylogenetic analysis resulted in three clusters (Figure 2); individuals located within these clusters were consistent with those grouped by population structure analysis. We used XLSTAT software v2019 (Addinsoft, Paris, France) to test the significance of the geographic distribution of 3 clusterings of proso millet through the *chi*-square test, and verify the results of the *chi*-square test with Fisher’s exact test. The results of the *chi*-square test showed that the overall *p*-value was <0.0001, which was lower than the significance level of 0.05. Therefore, we reject the null hypothesis that cluster and geographic distribution are independent, and the risk of error is less than 0.0001. Fisher’s exact test results that 17 of the 18 *p*-values are lower than the significance level 0.05. Fisher’s exact test also leads to a rejection of the null hypothesis. The results shows that the geographical distribution of 3 clusters of proso millet is significant.

### 2.3. Geographical Distributions and Agronomic Characteristics of Subclusters

To analyze the phylogenetic relationship of 578 proso millets, we did a phylogenetic analysis of the population and marked the country of origin (Figure 2). According to the delta K in the population structure analysis, three clusters A, B, and C were marked on the phylogenetic tree. However, in cluster A, it can be observed that the origin of a branch is more diverse than other accessions. In cluster C, the origins of accessions of a branch were located in plateau regions and high latitude regions. This show that the population can be subdivided based on the three subgroups. Determine the optimal number of subpopulations according to the delta K value calculated by the Structural Harvester program. Should be further divided into 10 subgroups. However, when K = 10 in population structure analysis, individuals of the same subgroup cannot be clustered in a cluster on the phylogenetic tree (Appendix A). Similarly, when K = 8, the same phenomenon occurred. However, when K = 6, individuals in the same subgroup can gather into clusters on the phylogenetic tree. We analyzed the agronomic traits data of 6 clusters and found that some traits have significant differences between clusters. 

In cluster A, branch A1 contained 174 germplasms; 98% of individuals were from South Korea. Branch A2 was distributed throughout Asia (Figure 2). Comparison of the agronomic traits of the two branches showed that the heading and harvest dates of branch A2 were earlier (heading = 27.13 days after sowing on average) than those of branch A1 (4.86 days later than A2) (Table 4). The number of branches per plant, panicle length, panicle width, stem diameter, and plant height were all lower in branch A2 than branch A1, with the most significant difference observed in plant height.

In cluster B, branch B1 showed high distribution frequency in Europe, whereas branch B2 was widely distributed in Eurasia. The heading and harvest dates of branch B1 were earlier than those of branch B2. The number of branches per plant, panicle length, panicle width, stem diameter, and plant height were lower in branch B1 than in branch B2.

In cluster C, branch C2, which included 94 germplasms, was distributed in high-latitude and high-altitude regions, whereas branch C1 was widely distributed in Eurasia. The heading date of branch C2 (27.06 days after sowing) was earlier than that of branch C1 (30.11 days after sowing); however, the harvest time of branch C2 was the latest among all branches (98.8 days after sowing). Thus, the germplasms of branch C2 have longer reproductive growth periods. Panicle length, panicle width, stem diameter, and plant height were lower in branch C2 than in branch C1. The detailed agronomic trait data are provided in Appendix A.

Therefore, we combined SSR marker, geographic distribution, and trait data to divide the 578 genetic resources into six clusters, such that clusters A1, A2, B1, B2, C1 and C2 were renamed as clusters 1–6, respectively.

The population structure analysis showed two distinct subpopulations for K = 2: cluster 5, which was widely distributed but mainly found in East and Southeast Asia, and cluster 4, which was also widely distributed (Figure 3 and Appendix A). At K = 3, cluster 5 was distinct from cluster 1, which is mainly distributed in Korea. At K = 4, cluster 5 was distinct from cluster 6, which was distributed in high-latitude and high-altitude regions such as Mongolia, Russia, and Ukraine. At K = 5, clusters 4 and 1 became distinct from cluster 2, which was distributed in Asia. At K = 6, cluster 3, which was distributed in Ukraine, Russia, and France, became distinct from cluster 5. We also performed a significant analysis on the geographic distribution of 6 clusters proso millet, and the results showed that the geographic distribution of 6 clusters proso millet was also significant. And because the higher the *chi*-square statistic, the lower the *p*-value, the geographic distribution of 6 clusters proso millet is more significant than that of 3 clusters proso millet.

As the number of subgroups (K) increases (K = 2 to K = 6), the population structure changes show that: cluster 1 is from cluster 5; cluster 6 is from cluster 5 and cluster 1; cluster 2 is from cluster 1 and cluster 4; cluster 3 is from cluster 5, cluster 1, and cluster 6 (Appendix A).

### 2.4. Gene Migration Analysis

Through comparing the migration rate (M) analysis results between the 6 clusters, we found the asymmetry of the migration rate between each two clusters. The direction of gene flow is considered to flow from clusters with high migration rates to clusters with low migration rates. The results showed that C5–C1 (114.845) > C1–C5 (64.223); C5–C6 (339.41) > C6–C5 (62.839); C1–C2 (249.636) > C2–C1 (90.07); C4–C2 (203.046) > C2–C4 (130.858); and C5–C3 (179.657) > C3–C5 (72.887) (Appendix A). The population structural analysis showed that as K increased, the changing trend in population structure remained consistent with the direction of gene flow among the six clusters (Appendix A).

We analyzed the gene flow direction from the origin of each subpopulation according to gene flow asymmetry; the detailed results are shown in Figure 4.

### 2.5. Evaluation of Antioxidant Potentials

The results of our antioxidant potential analysis showed that TPC ranged from 0 to 40.40 (median, 13.17) (Figure 5a). The three accessions with the lowest TPC content had polyphenol content values of 0.05 μg/g (128, IT33456), 0.51 μg/g (498, IT153558), and 0.52 μg/g (108, IT185556), whereas those with the highest TPC had polyphenol content values of 48.20 μg/g (281, IT100270) and 45.08 μg/g (112, IT212088).

The superoxide dismutase (SOD) activity ratio ranged from 34.78% to 107.45% (median, 71.56%). The four highest activity ratios were 107.45% (564, IT199344), 106.42% (173, IT108728), 105.9% (386, IT123985), and 105.9% (544, IT153553). Among the 14 extremely low SOD activity ratios, which indicated low SOD activity, the three lowest were 2.94% (64, IT175898), 5.36% (180, IT123951), and 9.32% (282, IT123971).

We performed Pearson’s correlation analysis of TPC and SOD (Figure 5b). The correlation coefficient was 0.3, indicating a moderately positive correlation. The significance level alpha was set to 0.05, *p* < 0.0001, and the correlation coefficient is significant. The r^2^ value indicated that the TPC of proso millet accounted for only 12% of the total variance in antioxidant activity. Differences in the TPC or content of other components may have had a greater impact on antioxidant activity.

### 2.6. Association Analysis

The results showed high-level linkage disequilibrium (LD). Among all 53 SSR marker loci, a total of 666 loci pairs were detected in the population. The significance threshold is set to *p* < 0.001. There are 108 pairs with significant and squared coefficient of correlation (r^2^) > 0.01, 80 pairs with r^2^ > 0.05, and 45 pairs with r^2^ > 0.1. There are 5 pairs with r^2^ > 0.5, including SSR-109 and SSR-120, SSR-70 and SSR-120, SSR-82 and SSR-331, SSR-70 and SSR-109, SSR-67 and SSR-82. 

The association analysis results showed that TPC was significantly associated with SSR-31 (*p* = 1.88E-04). SSR-31 explained 7.1% of the total phenotypic variation, comprising three genotypes. The average TPC values of three genotypes of each SSR are listed in Table 5.

## 3. Discussion

### 3.1. Gene Flow and Geographic Distributions

The gene flow analysis results did not identify a single domestication center for proso millet, as cluster 4 did not clearly originate from cluster 5. Our results could reflect either single or multiple domestication centers, as population structural analysis cannot assume that K = 1 [9]. Our gene flow analysis results were insufficient to demonstrate that cluster 4 originated from cluster 5; they only showed that gene flow from cluster 5 to cluster 4 was greater than that from cluster 4 to cluster 5. According to the local diversity of each cluster (Appendix A) and the gene flow analysis results, we determined the primary or secondary center of origin of each clusters. Nepal (and its surrounding areas) was the secondary center of origin for cluster 6. Our phylogenetic, population structure, and gene flow analysis results indicated that cluster 6 originated from cluster 5, and that the secondary center of origin was in Nepal and its surrounding areas, in high-latitude and high-altitude areas. This is similar to the three gene microcenters detected in Turkey, where the cultivation environment was exceptionally well-suited to wheat domestication [24]. In South Korea, cluster 1 was found to have derived from cluster 5, which is a unique germplasm resource. A previous study also identified Japan as an origin point for proso millet cluster 1 [9].

Cluster 1 is considered to be a unique proso millet germplasm resource that was domesticated in South Korea. Although one germplasm resource was collected in Russia, and one in Thailand, while two germplasms were collected in India, gene flow analysis showed that they were all derived from the Korean germplasm. Cluster 2 is distributed only in Asia. Gene flow analysis showed that these two clusters were introduced from China to South Korea, Mongolia, Uzbekistan, and India, and then from India to Russia and Thailand. Cluster 3 is widely distributed in Eurasia, with a diversity index of 0.143174. However, our gene flow results showed that cluster 3 originated in the fertile crescent of Turkey and then spread to France and India, from which it subsequently spread to Central Asia (Uzbekistan and Turkey) and finally to East Asia. In cluster 4, the highest genetic diversity was detected in germplasms native to Turkey; however, gene flow analysis showed that Turkey’s seed resources migrated from China to Uzbekistan, and then to Turkey. Cluster 4 originated in China and was introduced via Mongolia to Russia and North Korea. Germplasms of China and Mongolia also spread to Uzbekistan, and then from Uzbekistan to India, India to Thailand, Uzbekistan to Turkey, and Turkey to Ukraine. Cluster 5 occurred frequently in East, Southeast, and South Asia, and evolved according to a network pattern. Cluster 5 had the highest diversity; gene flow analysis pointed to China as the center of domestication, from which it migrated to Europe (Ukraine) via different routes, and then to Mongolia and finally back to China. Cluster 6 showed high abundance in high-latitude and high-altitude regions; it originated in Nepal and was then introduced into Ukraine, Russia, and Mongolia, and then into South Korea from Ukraine.

By combining the phylogeny, diversity, population structure, M, and gene flow analysis results, we obtained expansion paths for each cluster of proso millet that were consistent with a previously established archaeological map of the agricultural origins and migration of Neolithic and formative cultures [25]. In this study, we provided SSR markers combination of each cluster to identify the cluster where an individual is located (Appendix A). The highest diversity was found in clusters 4 and 5, and the longest branches (indicating the longest genetic distance from other germplasms) were found in clusters 3 and 5. Regions with high diversity were not identified as regions of origin, likely because gene introgression from other proso millet clusters resulted in high diversity. In a future study, we will amplify and sequence these genotypes to identify the markers with the greatest diversity (SSR-203, SSR-232, or SSR-365) as plant DNA barcodes, to detect other known antioxidants. Then, we will use these markers to accurately determine genotype composition and compare it with a neutral model to shed light on the population expansion associated with bottleneck events.

In this study, the population genetic diversity is low, and the genetic differentiation is high. We believe that the reason is that the proso millet is an inbred plant. Self-crossing reduces the effective population size and effective recombination rate. Compared with outcrossing, it directly leads to a decrease in polymorphism and an increase in linkage disequilibrium [26]. The increase in isolation between populations also directly stems from selfing or indirectly from evolutionary changes, leading to greater differentiation of molecular markers than during outcrossing. The lower effective recombination rate increases the possibility of free-riding and further reduces the internal diversity of inbreds, thereby increasing their genetic differentiation [27].

In addition, in the process of agronomic traits data collation, we observed that the standard deviation within each sub-cluster was very high (Table 4). The degradation of cultivars of proso millet to wild species may be the main reason for the large standard deviation. In this study, the population of 578 accessions is composed of wild species and landraces. Landraces degenerate into wild species, some genotypes are preserved, and may still be clustered with local landraces, but agronomic traits show differences. Feral derivatives of crop varieties may show a similar phenotype to that of the crop ancestor [28].

### 3.2. Association Analysis

The results showed high-level LD. This may be due to the mating system (selfing) of proso millet affecting the pattern of LD [29]. We arranged the TPC and SOD data corresponding to each accession on the outer circle with a simple bar (Appendix A). We observed that two red clades had higher TPCs, with average values of 25.4 and 23.2 μg/g. Blue clades had lower TPCs, with average values of 25.4 and 23.2 μg, respectively. Each clade can be clustered together because some SSR markers have the same amplification length. Therefore, we speculate that certain markers may be linked to the quantitative trait loci (QTL) associated with TPC. Comparison of the genotypes in these 4 clades showed high TPC in the 261-bp SSR-195 marker, and low TPC in the 269-bp SSR-195 marker (Appendix A). To further explore the relationships between these phenotypes and markers, we performed an association analysis of the phenotype and genotype data. Because proso millet is a selfed species, in the results of genetic differentiation analysis, a high degree of differentiation was observed in the population. This may lead to false positives in the association results. We choose to use the mixed linear model (MLM) for correlation analysis. In the MLM (Q + K) model, the population structure matrix (Q) and the kinship matrix (K) are used as random effects to control the false positives. SSR-31 associated with TPC explained >7.1% of the variance in TPC, indicating that there may be QTLs on both sides of the marker. 

There have been a large number of reports on the antioxidant mechanism of total phenols. The number and position of hydroxyl groups in phenolic compounds, and the nature of the substituents on the aromatic ring, determine the antioxidant capacity of plant extracts [30]. Whole-genome sequencing of the proso millet genome has been completed [31]. In future research, we will map the positions of SSR-31 identified in this study on the proso millet genome and develop flanking markers to gradually narrow the target range and determine the main proso millet genes influencing TPC and oxidation capacity, and clarify the types of phenolic compounds and antioxidant mechanisms in this species. This will provide new opportunities for high-quality proso millet breeding.

## 4. Materials and Methods

### 4.1. Plant Materials and Genotyping

In this study, Among 578 proso millet accessions collected in 17 countries of origin were obtained from the National Agro-biodiversity Center of the Rural Development Administration (RDA), Republic of Korea (http://genebank.rda.go.kr, accessed on 5 October 2021). The leaf tissue was sampled 5 weeks after sowing. Total DNA was extracted using the DNeasy® Plant Mini kit (Qiagen, Valencia, CA, USA) according to the manufacturer’s instructions. The DNA extraction used freeze-dried leaf tissue. The DNA was dissolved into 100 μL of water. Genomic DNA was quantified using a Nanodrop / UVS-99 instrument (ACTGene, Piscataway, NJ, USA) and determined the ratio of A260/A280 nm and dilute with water to 100 ng/μL. DNA quality was verified on 1% agarose gel. Store at −20 °C.

In a preliminary experiment, eight germplasms collected in regions with the largest differences in agronomic traits were selected to test the polymorphism of the marker-amplified products. We trialed 481 EST-SSR markers developed by Ali et al. [32]. These markers were originally used for the genetic diversity analysis of little millet (*Panicum sumatrense*). Among 481 little millet EST-SSR markers, 37 SSR markers with polymorphic amplification products were screened out (Table 6). The SSR makers were amplified in a total volume of 20 µL containing 50 ng of genomic DNA, 2 µL of each EST-SSR primer (10 pmol), 4 µL of 5× reaction buffer (Inclone Co.), 1 U of Taq DNA polymerase (Inclone Co.), 1.6 µL of dNTP (2.5 mM), and 11 µL of nuclease-free water. The amplification program was executed on GeneTouch thermal cycler (Bioer, Zhejiang, China) under the following conditions: Initial denaturation at 95 °C for 5 min, followed by 35 cycles at 95 °C for 40 s, 30 s at the annealing temperature, 30 s at 72 °C, and a final 10 min extension at 72 °C. The PCR products were separated using Fragment Analyzer™ 96 (Advanced Analytical Techologies, Ankeny, IA, USA) and the results were digitized using PROSize 3.0 software (Advanced Analytical Technologies, Ankeny, IA, USA).

### 4.2. Genetic Diversity Analysis

We used PowerMarker v3.25 software and PopGene32 v1.31 software to calculate the following genetic diversity indicators: observed Na, Ho, Nei’s gene diversity (H), I, and polymorphism information content (PIC). Geographic differences were evaluated using PowerMarker software following the estimation of Fst between geographic regions.

### 4.3. Population Structure Analysis

We used Structure software [33] to calculate the population structure for values of K ranging from 2 to 20, and used SIMCA v14.1 software (Umetrics, Umeå, Sweden) to perform orthogonal partial least-squares discrimination analysis (OPLS-DA) on 7 agronomic traits (number of branches per plant, heading date, harvest time, panicle length, panicle width, stem diameter, plant height (Appendix A)). Structure Harvester software [23] was used to calculate the most likely number of subspecies. Visualization was performed using CLUMPAK software [34]. 

### 4.4. Phylogenetic Analysis

We used 37 pairs of polymorphic SSR marker-amplified product length data for phylogenetic analysis, performed with PowerMarker software. MEGA X v10.0.4 (Kumar et al. 2018) software was used to draw a phylogenetic tree based on the 37 SSR markers amplified products size data of 578 individuals by the neighbor-joining method. The tree file was imported in nwk format into iTOL v5 [35], which was used to further visualize the phylogenetic tree. We used *chi*-square test and Fisher’s exact test in XLSTAT software v2019 (Addinsoft, Paris, France) to analyze the significance of clustering results and geographic distribution.

### 4.5. Effective Population Size and Migration

Migrate-n v4.4.3 software [36] was used to estimate gene flow. The data were entered into a microsatellite data model, the sampling increment was set to 1000, and the number of steps in the chain was set to 5000. We used a Brownian motion model to calculate theta values and effective mobility in two directions, and Bayesian analysis to calculate posterior distribution. Run model 2 to analyze the gene flow between six clusters, and gene flow between the origins of each cluster was then analyzed by run model 1.

### 4.6. Extraction and Determination of TPC

Phenolic compounds were extracted using an Ase-200 Accelerated Solvent Extractor (Dionex Corp., Sunnyvale, CA, USA). Briefly, 1.5 g of lyophilized and ground whole seed sample was mixed with 37.5 mL of 70% methanol, and the mixture was subjected to extraction for 40 min. Then, we centrifuged the sample at 4500 rpm for 10 min and the supernatant was collected. The obtained total phenol extract was stored at 4 °C.

TPC was determined using the Ainsworth colorimetric method and Folin–Ciocalteu reagent [37]. Initially, 0.2 mL of the sample extract was mixed with 2 mL of 2% Na_2_CO_3_ solution, and the mixture was incubated for 2 min at room temperature. Then, 0.2 mL of Folin–Ciocalteu reagent was added, and the absorbance was recorded at 750 nm after 30 min of incubation at room temperature. Known concentrations of gallic acid (20–100 ppm) were used to establish a standard curve and TPC was determined as μg GAE/g of dried seed weight from triplicate measurements.

### 4.7. SOD Activity Measurement

SOD activity was determined using the EZ-SOD Assay Kit (DoGenBio Co., Ltd., Seoul, Korea) following the improved Marklund method [38]. Initially, 1 mL of a 2-(4-iodophenyl)-3-(4-nitrophenyl)-5-(2,4-disulfophenyl)-2H-tetrazolium monosodium salt (WST) solution was mixed with 19 mL of buffer solution to prepare a WST working solution, and 15 µL of xanthine oxidase was mixed with 2.5 mL of dilution buffer to prepare an enzyme working solution. During analysis, 20 µL of sample solution was mixed with 200 µL of WST and 20 µL of enzyme working solution in 96-well plates. Then, the plates were incubated for 20 min at 37 °C and absorbance was measured at 450 nm using an Eon microplate spectrophotometer (BioTek Instruments, Inc., Winooski, VT, USA). SOD activity (inhibition rate) was calculated using the following equation:SOD activity = [(OD_blank1_ − OD_blank3_) − (OD_sample_ − OD_blank2_)] × 100/(OD_blank1_ − OD_blank3_)
where OD is the absorbance, blank 1 contains distilled and deionized water (ddH_2_O) instead of sample solution, blank 2 contains dilution buffer instead of enzyme solution, and blank 3 contains dilution buffer and ddH_2_O instead of enzyme and sample solution.

### 4.8. Association Analysis

Run packages (“corrplot”) in R v4.1.0 to analyze the correlation between TPC and SOD. The LD squared allele-frequency correlations (r^2^) was analyzed based on 1000 permutations using TASSEL v5.2.64 [39]. To identify SSR markers related to the target traits, we performed an association analysis of 37 SSR markers containing 177 proso millet genotypes with two sets of phenotypic data (TPC and SOD). However, detected associations in highly differentiated highly selfed species may be false positives. This may be due to the population structure and the existence of kinship between accessions. In the case of large differences in phenotypic frequency between different clusters, it is possible that the markers are only associated with clusters, and not associated with the quantitative trait loci. This analysis was performed using a mixed linear model (MLM) in TASSEL software [39]. MLM (Q + K) to control both population structure (Q) and kinship (K), to avoid false-positive results. Determine the population structure (Q) based on the population structure analysis final results. Kinship matrix (K) is provided by kinship analysis in Tassel. Markers were considered to be related to a trait if *p* < 0.01.

## 5. Conclusions

In this study, 37 SSR markers selected were used to genotyping 578 accessions of proso millet. The population shows a low level of diversity and high genetic differentiation. In population structure analysis, 578 proso millet accessions were divided into 3 clusters. Combining geographic distribution and trait data to distinguish more significance 6 clusters. Based on gene migration analysis, the formation process of the 6 clusters was estimated. Similarly, based on the asymmetry of the migration volume, the propagation paths of the 6 clusters in Asia and Europe are estimated. The accessions on cluster 1 are unique to Korea. Turkey may be the secondary center of origin and domestication of this genotype (cluster 3). We also found a cluster domesticated in Nepal (cluster 6), adapted to high latitude and high altitude cultivation conditions. We also studied the total phenolic content (TPC) and superoxide dismutase (SOD) activity and used SSR markers for correlation analysis. SSR-31 interpretation of TPC variables was 7.1%.

## Figures and Tables

**Figure 1 plants-10-02112-f001:**
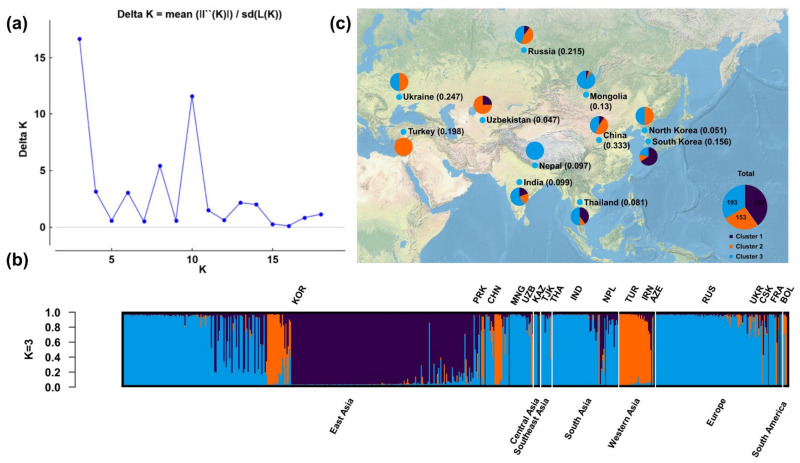
Population structural analysis results. (**a**) Determination of the optimal number of subpopulations (K) according to the values of ΔK calculated by the Structural Harvester program. (**b**) Corresponding population structure diagram and (**c**) geographical distribution for K = 3.

**Figure 2 plants-10-02112-f002:**
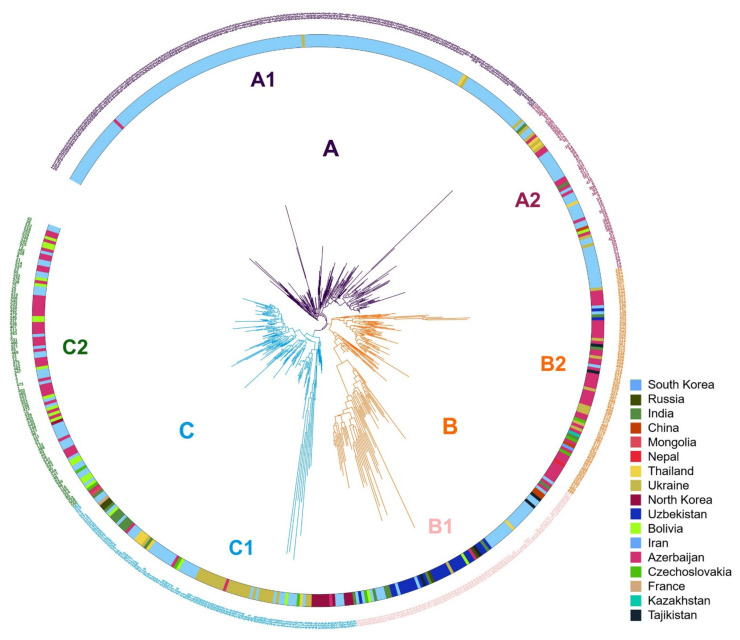
Phylogenetic tree for the three subgroups. Three clusters (A–C, indicated on branches) were identified, comprising a total of six subclusters with distinct geographical distributions and agronomic traits.

**Figure 3 plants-10-02112-f003:**
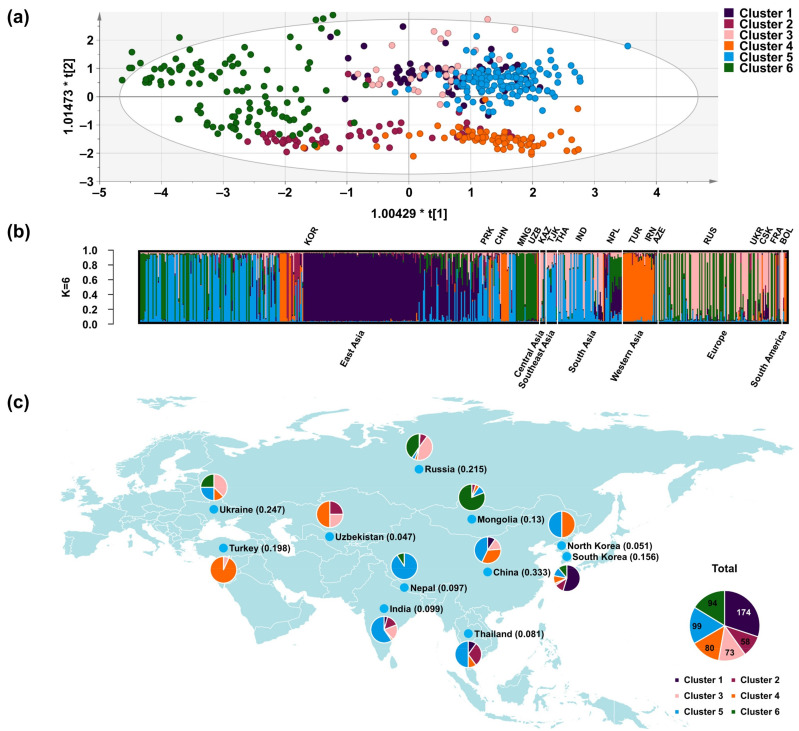
Population structural analysis and orthogonal partial least-squares discrimination analysis (OPLS-DA) plot. (**a**) OPLS-DA plot of six clusters. (**b**) Population structure plot for K = 6. (**c**) Geographical distribution of Eurasian germplasm showing the proportions of the six clusters in each region of origin, inferred by phylogenetic analysis.

**Figure 4 plants-10-02112-f004:**
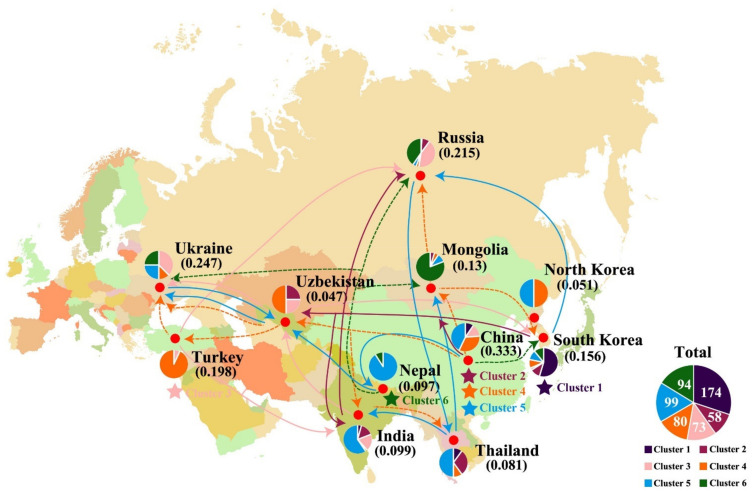
Geographical distribution and migration routes of the six proso millet clusters.

**Figure 5 plants-10-02112-f005:**
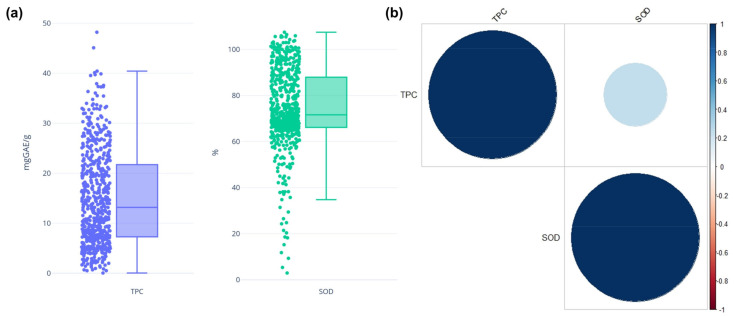
Distribution and correlation of total phenolic content and superoxide dismutase (SOD) activity among the 578 accessions used in this study. (**a**) Box-plot of TPC and SOD data. (**b**) Correlation plot of TPC and SOD.

**Table 1 plants-10-02112-t001:** Diversity information provided by 37 single-sequence repeat (SSR) markers.

Locus	Ng ^a^	Na ^b^	I ^c^	Ho ^d^	H ^e^	Fst ^f^	PIC ^g^	MAF ^h^
SSR-31	3	3	0.83	0	0.5185	0.5083	0.4277	0.5952
SSR-67	5	5	0.3864	0	0.1568	0.5596	0.1527	0.917
SSR-70	4	4	0.269	0	0.1208	0.211	0.1154	0.936
SSR-71	3	3	0.1518	0	0.0575	0.1335	0.0566	0.9706
SSR-82	6	6	0.4553	0	0.1926	0.4387	0.1857	0.8962
SSR-85	3	3	0.1527	0	0.0606	0.0957	0.0593	0.9689
SSR-86	2	2	0.1387	0	0.0603	0.119	0.0585	0.9689
SSR-92	3	3	0.5072	0	0.2864	0.6762	0.2534	0.8304
SSR-100	4	4	0.313	0	0.1341	0.0523	0.1294	0.9291
SSR-109	3	3	0.2646	0	0.1237	0.1991	0.1175	0.9343
SSR-120	5	5	0.6157	0	0.2895	0.4262	0.2723	0.8356
SSR-121	2	2	0.0803	0	0.0307	0.0705	0.0302	0.9844
SSR-127	3	3	0.1354	0	0.0508	0.0765	0.05	0.974
SSR-128	3	3	0.3346	0	0.1809	0.6783	0.1651	0.8997
SSR-129	3	3	0.2078	0	0.0837	0.0707	0.0819	0.9567
SSR-131	5	5	0.1612	0	0.0544	0.1865	0.0539	0.9723
SSR-142	3	3	0.1067	0	0.0375	0.1257	0.0371	0.981
SSR-143	5	5	0.0898	0	0.0274	0.0521	0.0273	0.9862
SSR-144	8	7	0.3365	0.0017	0.1199	0.5294	0.12	0.9369
SSR-146	4	4	0.0948	0	0.0308	0.0724	0.0306	0.9844
SSR-182	3	3	0.2275	0	0.0934	0.0609	0.0912	0.9516
SSR-195	3	3	0.5953	0	0.3512	0.6687	0.3024	0.7803
SSR-203	8	7	1.0966	0.0017	0.5816	0.4101	0.5086	0.545
SSR-232	17	6	1.2808	0.0657	0.6305	0.5055	0.5944	0.5536
SSR-331	4	4	0.5478	0	0.2672	0.4275	0.2511	0.8495
SSR-357	8	5	0.3716	0.0052	0.1535	0.3719	0.1491	0.9187
SSR-365	7	7	1.436	0	0.7331	0.4282	0.6884	0.3789
SSR-384	10	6	0.5456	0.0104	0.2291	0.2318	0.2178	0.878
SSR-386	6	6	0.1698	0.0035	0.0641	0.0604	0.0648	0.9663
SSR-394	4	4	0.7938	0.0017	0.52	0.5587	0.4107	0.5433
SSR-404	5	4	0.1716	0.0017	0.0672	0.3127	0.0642	0.9663
SSR-409	4	3	0.1446	0.0104	0.0636	0.0642	0.062	0.9671
SSR-420	4	3	0.0873	0.0087	0.034	0.0575	0.0287	0.9853
SSR-430	4	4	0.8734	0	0.4998	0.3517	0.44	0.6574
SSR-448	3	3	0.0885	0	0.0307	0.047	0.0304	0.9844
SSR-458	5	4	0.1054	0.0035	0.0375	0.0452	0.0371	0.981
SSR-460	5	3	0.1505	0.0035	0.0575	0.0742	0.0533	0.9723
Mean	4.7838	4.0811	0.387	0.0032	0.19	0.4545	0.1735	0.874
St. Dev	2.7226	1.3829	0.3434	0.0108	0.1954	0.1928	0.1705	0.1561

^a^ Number of genotypes in which each locus amplified alleles; ^b^ observed number of alleles; ^c^ Shannon’s information index; ^d^ observed heterozygosity; ^e^ Nei’s (1973) gene diversity; ^f^
*F*-statistic value for evaluation of geographical differentiation; ^g^ polymorphism information content; and ^h^ major allele frequency.

**Table 2 plants-10-02112-t002:** Genetic diversity information of each origin accession.

Origin	Ng ^a^	Na ^b^	I ^c^	Ho ^d^	H ^e^	PIC ^f^	MAF ^g^
South Korea	3.919 ± 1.402	3.73 ± 1.223	0.312 ± 0.338	0.002 ± 0.006	0.156 ± 0.192	0.145 ± 0.156	0.892 ± 0.154
North Korea	1.486 ± 0.683	1.486 ± 0.683	0.077 ± 0.231	0 ± 0	0.051 ± 0.149	0.148 ± 0.201	0.858 ± 0.197
China	3.108 ± 0.98	3.054 ± 0.928	0.188 ± 0.298	0.004 ± 0.013	0.333 ± 0.147	0.299 ± 0.13	0.857 ± 0.151
Mongolia	2.324 ± 1.275	2.27 ± 1.106	0.248 ± 0.254	0.005 ± 0.018	0.13 ± 0.14	0.151 ± 0.175	0.893 ± 0.155
Uzbekistan	1.243 ± 0.488	1.243 ± 0.488	0.068 ± 0.196	0.007 ± 0.041	0.047 ± 0.137	0.079 ± 0.153	0.912 ± 0.176
Thailand	1.541 ± 0.682	1.541 ± 0.682	0.142 ± 0.338	0.003 ± 0.016	0.081 ± 0.191	0.103 ± 0.134	0.92 ± 0.124
India	1.973 ± 1.174	1.919 ± 1.075	0.198 ± 0.338	0.002 ± 0.008	0.099 ± 0.18	0.094 ± 0.138	0.932 ± 0.116
Nepal	1.27 ± 0.684	1.243 ± 0.633	0.174 ± 0.353	0.007 ± 0.044	0.097 ± 0.196	0.044 ± 0.112	0.969 ± 0.079
Turkey	2.811 ± 1.135	2.811 ± 1.135	0.375 ± 0.409	0.004 ± 0.013	0.198 ± 0.225	0.255 ± 0.176	0.817 ± 0.146
Russia	2.892 ± 2.576	2.568 ± 1.516	0.417 ± 0.324	0.008 ± 0.031	0.215 ± 0.183	0.131 ± 0.193	0.892 ± 0.185
Ukraine	2 ± 1.115	1.973 ± 1.078	0.42 ± 0.436	0.01 ± 0.045	0.247 ± 0.247	0.198 ± 0.191	0.848 ± 0.164

^a^ Number of genotypes in which each locus amplified alleles; ^b^ observed number of alleles; ^c^ Shannon’s information index; ^d^ observed heterozygosity; ^e^ Nei’s (1973) gene diversity; ^f^ polymorphism information content; ^g^ major allele frequency.

**Table 3 plants-10-02112-t003:** Differentiation among origin populations according to the fixation index (Fst).

	AZE	BOL	CHN	CSK	FRA	IND	IRN	KAZ	KOR	MNG	NPL	PRK	RUS	THA	TJK	TUR	UKR	UZB
BOL	0.078 ***	-																
CHN	0.133 ***	0.16 ***	-															
CSK	0.189 ***	0.232 ***	0.104 ***	-														
FRA	0.081 ***	0.124 ***	0.156 ***	0.162 ***	-													
IND	0.093 ***	0.114 ***	0.071 ***	0.115 ***	0.096 ***	-												
IRN	0.133 ***	0.179 ***	0.065 ***	0.098 ***	0.191 ***	0.087 ***	-											
KAZ	0.162 ***	0.205 ***	0.089 ***	0.027 ***	0.135 ***	0.092 ***	0.098 ***	-										
KOR	0.127 ***	0.143 ***	0.055 ***	0.114 ***	0.13 ***	0.038 ***	0.081 ***	0.089 ***	-									
MNG	0.135 ***	0.145 ***	0.088 ***	0.136 ***	0.132 ***	0.063 ***	0.127 ***	0.109 ***	0.041 ***	-								
NPL	0.165 ***	0.186 ***	0.121 ***	0.117 ***	0.145 ***	0.082 ***	0.126 ***	0.09 ***	0.065 ***	0.078 ***	-							
PRK	0.162 ***	0.206 ***	0.07 ***	0.096 ***	0.189 ***	0.11 ***	0.079 ***	0.086 ***	0.088 ***	0.126 ***	0.134 ***	-						
RUS	0.085 ***	0.094 ***	0.07 ***	0.128 ***	0.096 ***	0.025 ***	0.103 ***	0.104 ***	0.031 ***	0.032 ***	0.073 ***	0.109 ***	-					
THA	0.116 ***	0.122 ***	0.096 ***	0.125 ***	0.114 ***	0.045 ***	0.12 ***	0.098 ***	0.057 ***	0.063 ***	0.099 ***	0.111 ***	0.052 ***	-				
TJK	0.054 ***	0.105 ***	0.15 ***	0.216 ***	0.108 ***	0.121 ***	0.16 ***	0.189 ***	0.15 ***	0.158 ***	0.17 ***	0.198 ***	0.107 ***	0.158 ***	-			
TUR	0.208 ***	0.248 ***	0.1 ***	0.215 **	0.27 ***	0.172 ***	0.114 ***	0.226 ***	0.145 ***	0.181 ***	0.225 ***	0.133 ***	0.17 ***	0.207 ***	0.238 ***	-		
UKR	0.111 ***	0.146 ***	0.074 ***	0.143 ***	0.133 ***	0.067 ***	0.124 ***	0.118 ***	0.054 ***	0.047 ***	0.09 ***	0.118 ***	0.04 ***	0.08 ***	0.134 ***	0.159 ***	-	
UZB	0.06 ***	0.051 ***	0.11 ***	0.174 ***	0.13 ***	0.088 ***	0.116 ***	0.147 ***	0.094 ***	0.096 ***	0.117 ***	0.133 ***	0.06 ***	0.104 ***	0.097 ***	0.181 ***	0.102 ***	-
Pop18	0.341 ***	0.272 ***	0.253 ***	0.137 **	0.212 ***	0.111 **	0.131 ***	0.157 ***	0.313 ***	0.221 ***	0.313 ***	0.215 ***	0.119	0.307 ***	0.237 ***	0.324 ***	0.431 ***	0.101 ***

** *p* < 0.01; *** *p* < 0.001. Fst value is 0-0.05, the degree of genetic differentiation among populations is low; Between 0.05–0.15, there is a moderate degree of genetic differentiation among populations; Between 0.15–0.25, the degree of genetic differentiation among populations is high; Above 0.25, there is great genetic differentiation among populations.

**Table 4 plants-10-02112-t004:** Differences in agronomic traits within each subcluster branch.

No	Cluster	Sub-Cluster	NB/P ^a^	HD ^b^	HT ^c^	PL ^d^	PW ^e^	SD ^f^	PH ^g^
1	A	A1	8.74 ± 1.98	31.99 ± 3.95	93.84 ± 5.84	29.81 ± 3.91	8.34 ± 1.9	2.75 ± 0.39	106.55 ± 14.2
1		A2	7.73 ± 1.94	27.13 ± 5.6	92.56 ± 6.71	25.73 ± 7.56	7.3 ± 3.34	2.23 ± 0.67	84.02 ± 31.64
2	B	B1	7.58 ± 2.15	23.34 ± 5.23	93.34 ± 5.32	19 ± 6.37	5.14 ± 2.65	1.68 ± 0.63	54.76 ± 28.67
2		B2	8.16 ± 2.13	29.83 ± 5.27	94.78 ± 5.19	26.45 ± 6.02	7.93 ± 3.18	2.43 ± 0.57	92.78 ± 27.58
3	C	C1	8.1 ± 2.04	29.39 ± 5.93	94.09 ± 5.94	26.78 ± 7.28	7.1 ± 2.64	2.37 ± 0.68	89.06 ± 30.08
3		C2	8.71 ± 2.18	26.81 ± 3.79	99.32 ± 5.36	22.42 ± 5.51	4.76 ± 1.59	1.71 ± 0.63	80.4 ± 21.59

^a^ Number of branches per plant; ^b^ heading date (days after sowing); ^c^ harvest time (days after sowing); ^d^ panicle length (cm); ^e^ panicle width (cm); ^f^ stem diameter (cm); ^g^ plant height (cm).

**Table 5 plants-10-02112-t005:** Association analysis results for SSR markers: TPC and SOD activity.

Trait	Marker Name	*p*-Value	r^2^	Genotype	Count	Corresponding Value (Average)
TPC (μg/g)	SSR-31	1.88E-04	0.07084	278/278	30	10.12
				287/287	344	12.89
				297/297	204	19.41

**Table 6 plants-10-02112-t006:** Primer information of the 37 SSR markers used in this study.

Name	Forward Primer	Reverse Primer	Annealing Temp. (°C)
SSR-31	ACTTCCCTAGAGTTCCAGT	TTCTGAAACTGTTCTATTGG	45
SSR-67	ACTAGGTAATTACAGGGGAG	GGCATGTGGAGTAGTAGTAT	46
SSR-70	ACTCATCTGACAAACTATGG	ATAGAACTGTGTGTTGGTGT	45
SSR-71	ACTCATGATTAAAGGGTGAT	TGTGACAACATTGTGAATAG	46
SSR-82	ACCAGCCCCAACTAC	ATTGTTTATGTGATCTCAGG	45
SSR-85	ACCAGTACGGCAACC	ATTTCTCTTTGATCTTCTCC	45
SSR-86	ACCAGTACGGCAACC	TTGATCTTCTCCTTAATGC	45
SSR-92	ACCCACCCAACCAGT	TACTTTGTCCTTTTCCAGTA	46
SSR-100	ACCTAGACAAATGCGTACT	CAAAACCAAACCCTCTC	45
SSR-109	ACCTTAAGGATTGGAATATC	GTTGAGTAAGTTTCTCCTCA	46
SSR-120	ACGACCATGATCTCATAAC	GAGGATGATGAGTAGGAAGT	45
SSR-121	ACGACGATGATGATGAC	TCTGGTCAAGTACTCAATTC	46
SSR-127	ACGAGGAGATGGATCAG	CTCTCTGTCCGTGGTC	46
SSR-128	ACGATGATGAAGAAGCA	GAACTGGCAGAAGCAC	46
SSR-129	ACGATGGGGTCTACG	AGCTTAACCCTGAACTTCT	45
SSR-131	ACGCAGCCTCATCAT	TAAGAAGCTGAGATTTGGT	45
SSR-142	ACTAAGAGGAAGCCTATGTT	AACTGCAGCTACATTGTATT	45
SSR-143	ACTAAGAGGAAGCCTATGTT	TACAGCAGTGCAGATATTTA	45
SSR-144	ACTAAGAGGAAGCCTATGTT	TTAAGCTGGAAAGTAATCAG	45
SSR-146	ACTACAAGAGCAAGTCCAC	AAATACAACATTGCAAGACT	45
SSR-182	ACAACAGATTTCTAAACCAA	TCTCGGAGAACATCAAG	45
SSR-195	ACAAGTAATTTCCGTATCAA	AGTCAGAAGAGTCAACAACA	45
SSR-203	ACACAAACTTGATACTCTGG	GTGTTGTATGCAACTGAAG	45
SSR-232	ACAGTAATCTACGCAACAAT	ATTTTTCCCTTTTGTTCTAT	45
SSR-331	AAGCAGCTGAGGATAAAG	GTACACTCCGAACTCAAAG	45
SSR-357	AAGGTGATCATGTAATGAGA	GTGTCATATTGGCAGTAAGT	45
SSR-365	AAGTACGAGAACCTGATTG	AGTTTCTTACCCTTTTCAAC	53
SSR-384	AAGTTCAGCGACTTAAGATA	TGATATTGTCCTCAAATGAC	45
SSR-386	AAGTTTCTACCCTTTTCAAC	AAGTACGAGAACCTGATTG	53
SSR-394	AATAATCAACAACCGAATTA	CTCCTATCCATTACTGATGA	45
SSR-404	AAGAGAAAGAACGGCTATT	ACAGAGCTCACAATATGTTC	53
SSR-409	AAGAGTAGGAGACCCATTAC	AGGTAAAAATATGCCTGAAT	53
SSR-420	AAGAAGGGTAGTGATGGAT	TTGTTTTAGACTCTCCTCAA	53
SSR-430	AACTCTGTCATATGGTTACG	AGGGGATTCTTCAGATAAT	45
SSR-448	AAGAAATCAGAGAGGACAGT	ACAAGAAAAACTCGAGTACA	53
SSR-458	AACTACGTACAAAAATGGAA	CATAAATAGCGAGCATACAT	50
SSR-460	AACTAGCAATAGGTTGAACA	GACTGGTACATTTTCAAAGA	45

## Data Availability

Data is contained within the article or Appendix A.

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
