# Peer review of "Phylogeography and Antioxidant Activity of Proso Millet (Panicum miliaceum L.)"

_plants, 2021, doi:10.3390/plants10102112_

Round 1
Reviewer 1 Report
The submitted paper is about Phylogeography and antioxydant capacity of proso millett (Panicum miliaceum L.). Huge number of accessions, derived from different parts of the world, of proso millet were evaluated in the study. This paper contains lots of useful information about origin of the millet and the gene flow of the varieties, which are grown in the countries mentioned in the paper. Were the examined varieties bred or selected in the mentioned countries? Is it possible to attach a list about the examined varieties in this trial? The list can help a lot to understand better the results.
The data were evaluated by statistical programs, and the standard deviations were marked in the tables 2 and 4, but the homogenous groups aren’t indentified in the paper. Is it possible to enlarge the Figures 1 and 3, because those are not readable?
On the figure 5 the regession rate is very low. Is it possible to change the figure type to have better regession?
Author Response
First of all, we would like to thank the editor and reviewers for their efforts and suggestions on our manuscript, which have certainly helped to improve the quality of the manuscript. On the basis of the reviewer’s comments, we have revised our manuscript to improve its current quality. Changes / justifications to their comments and suggestions are provided below.
The submitted paper is about Phylogeography and antioxydant capacity of proso millett (Panicum miliaceum L.). Huge number of accessions, derived from different parts of the world, of proso millet were evaluated in the study. This paper contains lots of useful information about origin of the millet and the gene flow of the varieties, which are grown in the countries mentioned in the paper. Were the examined varieties bred or selected in the mentioned countries? Is it possible to attach a list about the examined varieties in this trial? The list can help a lot to understand better the results.
Author reply: Thank you very much for your comments. The 578 proso millet genetic resources are all from the National Agrobiodiversity Center in South Korea. Only the country of origin is provided on the record, and the more detailed sampling location (latitude and longitude) is not indicated.
We have previously neglected to point out the information location of germplasm resources (Supplementary dataset 1), and have made corresponding revisions in the manuscript.
1.1. The data were evaluated by statistical programs, and the standard deviations were marked in the tables 2 and 4, but the homogenous groups aren’t indentified in the paper. Is it possible to enlarge the Figures 1 and 3, because those are not readable?
Figures 1 and 3 have been enlarged.
Author reply: Thank you very much for your comments. We have observed that the standard deviations within each subcluster is high. The degradation of cultivars of millet to wild species may be the main reason for the large standard deviation. In this study, the population of 578 accessions is composed of wild species and landraces. Landraces degenerate into wild species, some genotypes are preserved, and may still be clustered with local landraces, but agronomic traits show differences. Feral derivatives of crop varieties may show a similar phenotype to that of the crop ancestor (Zhang, 2018).
Zhang, J., Lu, H., Liu, M., Diao, X., Shao, K., & Wu, N. (2018). Phytolith analysis for differentiating between broomcorn millet (Panicum miliaceum) and its weed/feral type (Panicum ruderale). Scientific reports, 8(1), 1-9.
1.2. On the figure 5 the regession rate is very low. Is it possible to change the figure type to have better regession?
Author reply: Thank you very much for your comments. We redo the correlation analysis between TPC and SOD by running packages ("corrplot") in R v4.1.0.
Wei, T., Simko, V., Levy, M., Xie, Y., Jin, Y., & Zemla, J. (2017). Package ‘corrplot’. Statistician, 56(316), e24.

Reviewer 2 Report
This is an interesting paper trying to study genetic diversity in highly selfing plant species. However, the authors do not mention, that Panicum miliaceum is selfed species. The fact that this species is selfed (actually highly inbred) should have direct consequences on the results of genetic diversity and genetic differentiation among populations, and this (biological features of the species) should be described in detail in the paper to make the context for the results.
I have several detailed comments which could be helpful in improving the manuscript. The manuscript needs a careful correction of English to make the text fully understandable without misleading.
Minor comments are included below.
105 – it is hard to say about the gene flow, if the species is self-fertile mostly, the term gene flow is used as the dispersal of genes by pollen or seed, or by hybridization; was gene flow used in that meaning?
345- where is the information about the details of landraces, do they form some type of groups or populations? What is the geographic origin of landraces?
350-351 what is the source of SSR markers used in the study, please cite the reference.
Were there any tests for linkage disequilibrium among the marker set?
354 – what type of instrument was used for PCR amplification, and what for fragment sizing?
359-362 – why Fis was not estimated in the study?
366 – provide a reference for SIMCA software. What type of data has been analyzed with that software?
371 – write ‘MEGA X’ – blank space needed before X, and provide software reference.
371 – was the phylogenetic tree made based on individuals?
373 – provide a reference for ITOL
116 – the problem of measuring heterozygosity in a selfing species is interesting and challenging. I do not think that the conclusion about the gene flow among individuals is appropriate at 100%. The question is to measure the levels of inbreeding in this species. It is possible and should be done. Also, it should be possible to test if the population is in H-W equilibrium, given the presence of inbreeding. Many generations of selfing should force almost exclusive homozygosity, which is not observed. This issue should be investigated in detail.
Another interesting aspect is to try to estimate the level of null alleles because the presence of null alleles makes the overestimation of inbreeding. I suggest trying INEST software described in Chybicki, I. J., & Burczyk, J. (2009). Simultaneous estimation of null alleles and inbreeding coefficients. J Hered, 100(1), 106-113. doi:10.1093/jhered/esn088. However, trying to estimate null alleles in the selfed species could be a problem because of such a high level of actual inbreeding
119 – genetic differentiation seems to be very high, and this is in line with predominant selfing mating system. Panicum miliaceum is know to be selfing species, which may promote genetic differentiation among populations. This should be discussed.
136 – a word ‘genomes’ seems inappropriate because this suggests that whole genomes were investigated. Usse accesions or a similar term
Table 3 provides measures for pairwise differentiation among origin populations.
And what is the global Fst for all populations?
Please test if there is a correlation between the level of Fst and the physical distance between populations. This could be simply done as a Mantel test, or just correlations and this way isolation by distance could be tested.
Is it possible to estimate the significance of Fst for each pair? Which estimates were significant?
Please explain why Fst for individual loci are so high (about 0.80) but for pairwise comparisons, they are rather small.
168 – but could you test the significance?
Figure 2. Please provide a better wuality figure so details could be read. This also refers to other figures.
172 – it should be clear, that the subclusters were defined arbitrarily. Someone else could define a different number of subclusters
Table 4 – it seems that the variance within each subcluster is high and there are no significant differences between subclusters. Please comment on that.
220- what do you mean by genotype here? (M1-6) it is absolutely not clear. It seems they are subgroups or subclusters, but please be consistent in terminology.
219 – it is hard to mention gene flow analysis it is rather gene migration analysis.
246 – was correlation significant? At what level?
251 – the association analysis in selfed species is a specific issue. Please provide some background on this topic. Association may be caused not by linkage between marker and gene responsible for the quantitative trait, but due to the chance of fixation of marker locus and the QTL. Please provide some discussion on this topic.
Then detected associations in highly selfed species with high differentiation might be spurious – and misleading. The problem is that in selfed species the differentiation on quantitative traits might be correlated to one of the studied loci, but because of a specific pattern of genetic differentiation among selfed species at neutral loci, some of the loci might just follow the pattern of the quantitative traits by chance. This really needs careful assessment.
Author Response
First of all, we would like to thank the editor and reviewers for their efforts and suggestions on our manuscript, which have certainly helped to improve the quality of the manuscript. On the basis of the reviewer’s comments, we have revised our manuscript to improve its current quality. Changes / justifications to their comments and suggestions are provided below.
This is an interesting paper trying to study genetic diversity in highly selfing plant species. However, the authors do not mention, that Panicum miliaceum is selfed species. The fact that this species is selfed (actually highly inbred) should have direct consequences on the results of genetic diversity and genetic differentiation among populations, and this (biological features of the species) should be described in detail in the paper to make the context for the results.
Author reply: Thank you very much for your comments.
Add “The growth cycle is only 60 to 100 days, one panicle, the plant height ranges from 30 to 100 cm, with few tillers, and an adventitious root system (Habiyaremye, 2017).”
Habiyaremye, C., Matanguihan, J. B., D’Alpoim Guedes, J., Ganjyal, G. M., Whiteman, M. R., Kidwell, K. K., & Murphy, K. M. (2017). Proso millet (Panicum miliaceum L.) and its potential for cultivation in the Pacific Northwest, US: A review. Frontiers in plant science, 7, 1961.
Add “Proso millet is a self-bred plant. Selfing can lead to a decrease in genetic diversity. The existing self-bred plants all originated recently. The explanation given by Stebbins for this phenomenon is that selfing plants always appear, but selfing leads to the loss of population genetic diversity, which cannot adapt to environmental changes, and the probability of extinction is higher than other populations (Stebbins, 1957).”
Stebbins, G. L. 1957. Self-fertilization and population variability in the higher plants. Am. Nat. 91:337–354.

Reviewer 3 Report
Dear Authors,
The article Sadaf et al. describes “Phylogeography and antioxidant activity of proso millet (Panicum miliaceum L.)” The manuscript presented by the authors is interesting and introduces new elements. P. miliaceum is a very important economic plant. It is widely used in human and animal nutrition. The article was written very well and solidly, presenting the necessary genetic analysis. Moreover, it is very interesting to present the biological activity of the analyzed plant. It contains a few of minor errors that must be corrected in order for the article to be published in Plants MDPI.
1) I suggest not to repeat keywords in the title.
2) Please add a short summary chapter "Conclusion"
After applying all corrections recommend publishing in Plants MDPI.
Sincerely
Author Response
First of all, we would like to thank the editor and reviewers for their efforts and suggestions on our manuscript, which have certainly helped to improve the quality of the manuscript. On the basis of the reviewer’s comments, we have revised our manuscript to improve its current quality. Changes / justifications to their comments and suggestions are provided below.
The article Sadaf et al. describes “Phylogeography and antioxidant activity of proso millet (Panicum miliaceum L.)” The manuscript presented by the authors is interesting and introduces new elements. P. miliaceum is a very important economic plant. It is widely used in human and animal nutrition. The article was written very well and solidly, presenting the necessary genetic analysis. Moreover, it is very interesting to present the biological activity of the analyzed plant. It contains a few of minor errors that must be corrected in order for the article to be published in Plants MDPI.
3.1. I suggest not to repeat keywords in the title.
Author reply: Thank you very much for your comments. Replaced “Antioxidant activity; proso millet; genetic diversity; phylogeographic analysis; SSR marker; total phenol content” with “Gene migration; population structure; genetic diversity; association analysis; SSR marker; total phenol content”
2.2. Please add a short summary chapter "Conclusion"
Author reply: Thank you very much for your comments. In this study, 37 SSR markers selected were used to genotyping 578 accessions of proso millet. The population shows a low level of diversity and high genetic differentiation. In population structure analysis, 578 millet accessions were divided into 3 clusters. Combining geographic distribution and trait data to distinguish more significance 6 clusters. Based on gene migration analysis, the formation process of the 6 clusters was estimated. Similarly, based on the asymmetry of the migration volume, the propagation paths of the 6 clusters in Asia and Europe are estimated. Cluster 1 is accessions unique to Korea. Turkey may be the secondary center of origin and domestication of this genotype (cluster 3). We also found a cluster domesticated in Nepal (cluster 6), adapted to high latitude and high altitude cultivation conditions. We also studied the total phenol content (TPC) and superoxide dismutase (SOD) activity, and used SSR markers for correlation analysis. SSR-31 interpretation of TPC variables was 7.1%.

Reviewer 4 Report
The authors of the paper “Phylogeography and antioxidant activity of proso millet (Panicum miliaceum L.)” performed a study of the genetic diversity using SSR marker and an analysis of the antioxidant activity evaluating the total phenolic content (TPC) and superoxide dismutase (SOD) content of 578 millet accession from different regions of the world. The results highlighted the genetic structure of millet and gave new insight into its domestication history. Furthermore, the authors found SSR markers associated with agronomical traits such as TPC and SOD content. The paper can be published since the research was well conducted and the results are very interesting for the breeding of the millet. However, the paper needs to be improved before its publication. The results are not always well showed. The discussion section is not well written. It could be improved. Here are some answers and hints for the authors:
- In line 101, you should improve this sentence “…the TPC and antioxidant properties of 578 proso millet accessions collected after harvest in different regions of Korea,…” in the introduction section. It is better to highlight that the 578 accessions are from different parts of the world.
- In line 103 you stated that: “This study has increased the number of proso millet accessions and markers,….”. It is not properly right since the accessions and the markers were just known before your study.
- In line 59 there is a typo. There is an extra “that” in the following sentence “…suggests that that proso millet originated….”.
- At line 110, what do you mean for stable bands?
- At line 111, you might specify that your accessions are from different countries. This could help the reader.
- Recheck the grammar at line 119.
- In line 132, it should be better to use the word accessions instead of population.
- You did not describe the parameter MAF in the results section.
- In line 135, the sentence “The Fst and migration rate (M) were used to evaluate genetic flow among regions of 135 origins (Table 3)” is not properly right since you did not show the migration rate value in table 3.
- At line 182, introduce the units for the agronomical traits in the table.
- At line 200, you stated: “Therefore, we combined SSR marker, geographic distribution, and trait data to divide the 578 genetic resources into six clusters, such that clusters A1, A2, B1, B2, B5, and B6 were renamed as clusters 1–6, respectively”. How did you make this analysis? Did you perform the orthogonal partial least-squares discrimination analysis (OPLS-DA)?
- In line 202, you did not consider cluster C and its subclusters C1 and C2, did you? Why?
- It seems to me that there is a mistake in the sentences at lines 200-202. You wrote that you analyzed all 578 genetic resources but after you specify that consider all the clades A and B and their relative subclades excluding the C one. Recheck.
- At lines 215-218, you stated: “As the number of subgroups (K) increases (K=2 to K=6), the population structure 215 changes show that: Cluster 1 is from Cluster 5; Cluster 6 is from Cluster 5 and Cluster 1; 216 Cluster 2 is from Cluster 1 and Cluster 4; Cluster 3 is from Cluster 5, Cluster 1, and Cluster 217 6 (Figure S2)”. I don’t think that you can deduce that. Is this approach commonly used? It should be better to remove these sentences.
- At line 220, could you describe the results of the study of the asymmetry of the gene flow among genotypes?
- The paragraph “2.4. Gene flow analysis” should be improved. The results are not shown clearly.
- In line 220, It should be better to use the word “clusters” instead of “genotypes”.
- It should be better to improve the description of the legend in figure 5.
- At line 230, you might use the word clusters instead of genotypes.
- Use the same style to indicate the SSR marker. In line 253, you used the upper-case style for them meanwhile you used normal style in the previous part of the text.
- Could you use the results of the association study for the other remaining SSR markers?
- In this study, we identified non- polymorphic SSRs for each genotype and developed a classification system based on these SSR markers similar to the AFLP fingerprint analysis system (Table S2). Could you explain me better?
- The data showed in figure 6 are not presented in the results section as well as the relative analysis is not clearly described in the discussion section.
- Maybe figure 6 you could display in the supplementary methods.
- You did not describe how you extracted the genomic DNA in the material and methods section.
- How many nanograms did you use for the amplification?
- At line 350, in the material and methods section, could you specify how you chose the 481 EST-SSR markers. Did you retrieve it from the literature?
- In line 389 you should correct the typo for Na2CO3.

Author Response
First of all, we would like to thank the editor and reviewers for their efforts and suggestions on our manuscript, which have certainly helped to improve the quality of the manuscript. On the basis of the reviewer’s comments, we have revised our manuscript to improve its current quality. Changes / justifications to their comments and suggestions are provided below.
The authors of the paper “Phylogeography and antioxidant activity of proso millet (Panicum miliaceum L.)” performed a study of the genetic diversity using SSR marker and an analysis of the antioxidant activity evaluating the total phenolic content (TPC) and superoxide dismutase (SOD) content of 578 millet accession from different regions of the world. The results highlighted the genetic structure of millet and gave new insight into its domestication history. Furthermore, the authors found SSR markers associated with agronomical traits such as TPC and SOD content. The paper can be published since the research was well conducted and the results are very interesting for the breeding of the millet. However, the paper needs to be improved before its publication. The results are not always well showed. The discussion section is not well written. It could be improved. Here are some answers and hints for the authors:
1. In line 101, you should improve this sentence “…the TPC and antioxidant properties of 578 proso millet accessions collected after harvest in different regions of Korea,…” in the introduction section. It is better to highlight that the 578 accessions are from different parts of the world.
Author reply: Thank you very much for your comments. Replaced "the TPC and antioxidant properties of 578 proso millet accessions collected after harvest in different regions of Korea" with "the TPC and antioxidant properties of 578 proso millet accessions collected in 17 countries of origin of the world"
2. In line 103 you stated that: “This study has increased the number of proso millet accessions and markers,….”. It is not properly right since the accessions and the markers were just known before your study.
Author reply: Thank you very much for your comments. Replace "This study has increased the number of proso millet accessions and markers" with "This study has screened out 37 EST-SSR markers suitable for proso millet"
3. In line 59 there is a typo. There is an extra “that” in the following sentence “…suggests that that proso millet originated….”.
Author reply: Thank you very much for your comments. Corresponding changes have been made in the manuscript.
4. At line 110, what do you mean for stable bands?
Author reply: Thank you very much for your comments. What we mean is "screening out 37 EST-SSR markers that can be successfully amplified in the DNA of all individuals and have polymorphisms."
Corresponding changes have been made in the manuscript.
Replace "37 with stable bands and polymorphisms" with "37 EST-SSR markers that can be successfully amplified in the DNA of all individuals and have polymorphisms were screened out"
5. At line 111, you might specify that your accessions are from different countries. This could help the reader.
Author reply: Thank you very much for your comments. Replace "Among 578 millet accessions," with "Among 578 proso millet accessions collected in 17 countries of origin"
6. Recheck the grammar at line 119.
Author reply: Thank you very much for your comments. Replace "The fxation index (Fst), a genetic differentiation index, was calculated using 37 markers; values ranged from 0.2669 (ss1-365) to 0.9726 (ssr-143), with an average of 0.81." with "The fixation index (Fst) is used to measure the proso millet population genetic differentiation. Among the 37 SSR markers, each marker provided a different ability to distinguish genetic differentiation, ranging from 0.2669 (ssr-365) to 0.9726 (ssr-143), with an average of 0.81."
7. In line 132, it should be better to use the word accessions instead of population.
Author reply: Thank you very much for your comments. We replaced "population," with "accessions".
8. You did not describe the parameter MAF in the results section.
Author reply: Thank you very much for your comments. Add to the results "The average major allele frequency (MAF) was 0.8740, with a range of 0.3789 (SSR-365) to 0.9862 (SSR-143).”
9. In line 135, the sentence “The Fst and migration rate (M) were used to evaluate genetic flow among regions of 135 origins (Table 3)” is not properly right since you did not show the migration rate value in table 3.
Author reply: Thank you very much for your comments. Replace "The Fst and migration rate (M) were used to evaluate genetic flow among regions of 135 origins (Table 3)" with "Pairwise Fst is used to evaluate the degree of genetic differentiation among 17 countries of origin (Table 3). "
10. At line 182, introduce the units for the agronomical traits in the table.
Author reply: Thank you very much for your comments. Corresponding changes have been made in the manuscript.
11. At line 200, you stated: “Therefore, we combined SSR marker, geographic distribution, and trait data to divide the 578 genetic resources into six clusters, such that clusters A1, A2, B1, B2, B5, and B6 were renamed as clusters 1–6, respectively”. How did you make this analysis? Did you perform the orthogonal partial least-squares discrimination analysis (OPLS-DA)?
Author reply: Thank you very much for your comments. First, we performed orthogonal partial least-squares discrimination analysis (OPLS-DA) based on 7 agronomic traits data in 6 clusters. The results show that all 6 clusters can be distinguished. The purpose of ANOVA analysis is to compare the differences between clusters 1 and 2, 3 and 4, and 5 and 6. Clusters 1 and 2, 5 and 6, have strong geographic differences. The plant lengths in cluster 3 and 4 are significantly different.
12. In line 202, you did not consider cluster C and its subclusters C1 and C2, did you? Why?
Author reply: Thank you very much for your comments. Replaced "B5 and B6" with "C1 and C2".
13. It seems to me that there is a mistake in the sentences at lines 200-202. You wrote that you analyzed all 578 genetic resources but after you specify that consider all the clades A and B and their relative subclades excluding the C one. Recheck.
Author reply: Thank you very much for your comments. Corresponding revisions were made in the manuscript.
14. At lines 215-218, you stated: “As the number of subgroups (K) increases (K=2 to K=6), the population structure 215 changes show that: Cluster 1 is from Cluster 5; Cluster 6 is from Cluster 5 and Cluster 1; 216 Cluster 2 is from Cluster 1 and Cluster 4; Cluster 3 is from Cluster 5, Cluster 1, and Cluster 217 6 (Figure S2)”. I don’t think that you can deduce that. Is this approach commonly used? It should be better to remove these sentences.
Author reply: Thank you very much for your comments. Corresponding revisions were made in the manuscript.
15. At line 220, could you describe the results of the study of the asymmetry of the gene flow among genotypes?
Author reply: Thank you very much for your comments. Add "Through comparing the migration rate (M) analysis results between the 6 clusters, we found the asymmetry of the migration rate between each two clusters. The direction of gene flow is considered to flow from clusters with high migration rates to clusters with low migration rates."
16. The paragraph “2.4. Gene flow analysis” should be improved. The results are not shown clearly.
Author reply: Thank you very much for your comments. Corresponding changes have been made in the manuscript.
17. In line 220, It should be better to use the word “clusters” instead of “genotypes”.
Author reply: Thank you very much for your comments. Corresponding changes have been made in the manuscript.
18. It should be better to improve the description of the legend in figure 5.
Author reply: Thank you very much for your comments. Corresponding changes have been made in the manuscript.
19. At line 230, you might use the word clusters instead of genotypes.
Author reply: Thank you very much for your comments. Corresponding changes have been made in the manuscript.
20. Use the same style to indicate the SSR marker. In line 253, you used the upper-case style for them meanwhile you used normal style in the previous part of the text.
Author reply: Thank you very much for your comments. Uniformly changed to uppercase style.
21. Could you use the results of the association study for the other remaining SSR markers?
Author reply: Thank you very much for your comments. We performed an association analysis based on MLM (Q+K) using the amplified length of 37 SSR markers and two phenotype data. Only one significant correlation marker related to SSR-31 (P= 1.8822 × 10-4) explained 7.1% of TPC phenotypic variation. The remaining 36 SSR markers were not significantly associated with the two phenotypes.
22. In this study, we identified non- polymorphic SSRs for each genotype and developed a classification system based on these SSR markers similar to the AFLP fingerprint analysis system (Table S2). Could you explain me better?
Author reply: Thank you very much for your comments. Replaced "In this study, we identified non- polymorphic SSRs for each genotype and developed a classification system based on these SSR markers similar to the AFLP fingerprint analysis system (Table S2)." with "In this study, we provided SSR markers combination of each cluster for identify the cluster where an individual is located (Table S2)."
23. The data showed in figure 6 are not presented in the results section as well as the relative analysis is not clearly described in the discussion section.
Author reply: Thank you very much for your comments. Replace "Figure 6" with "Figure S3" and modify the description in the discussion section.
We arranged the TPC and SOD data corresponding to each accession on the outer circle with a simple bar. We observed that two red clades had higher TPCs, with average values of 25.4 μg/g and 23.2 μg/g. Blue clades had lower TPCs, with average values of 25.4 μg/g and 23.2 μg, respectively. Each clade can be clustered together because some SSR markers have the same amplification length. Therefore, we speculate these SSR markers may be linked to the quantitative trait loci (QTL) associated with TPC.
24. Maybe figure 6 you could display in the supplementary methods.
Author reply: Thank you very much for your comments. Replace "Figure 6" with "Figure S3".
25. You did not describe how you extracted the genomic DNA in the material and methods section.
Author reply: Thank you very much for your comments. The leaf tissue was sampled 5 weeks after sowing. Total DNA was extracted using the DNeasy® Plant Mini kit (Qiagen, Valencia, CA, USA) according to the manufacturer’s instructions. DNA extraction use freeze-dried leaf tissue. The DNA was dissolved into 100 μL of water. Genomic DNA was quantified using a Nanodrop / UVS-99 instrument (ACTGene, Piscataway, NJ, USA) and determined the ratio of A260 / A280 nm, and dilute with water to 100 ng/μL. DNA quality was verified on 1% agarose gel. Store at -20℃.
26. How many nanograms did you use for the amplification?
Author reply: Thank you very much for your comments. We use 50 nanograms of genomic DNA for the amplification.
Add “The SSR makers were amplified in a total volume of 20 μL containing 50 ng of genomic DNA, 2 μL of each EST-SSR primer (10 pmol), 4 μL of 5× reaction buffer (Inclone Co.), 1 U of Taq DNA polymerase (Inclone Co.), 1.6 μL of dNTP (2.5 mM), and 11 μL of nuclease-free water.”
27. At line 350, in the material and methods section, could you specify how you chose the 481 EST-SSR markers. Did you retrieve it from the literature?
Author reply: Thank you very much for your comments. Add “We trialled 481 EST-SSR markers developed by Ali et al. (Ali, 2017). These markers were originally used for genetic diversity analysis of little millet (Panicum sumatrense).”
Ali, A., Choi, Y. M., Hyun, D. Y., Lee, S., Kim, J. H., Oh, S., & Lee, M. C. (2017). Development of EST-SSRs and assessment of genetic diversity in little millet (Panicum sumatrense) germplasm. Korean Journal of Plant Resources, 30(3), 287-297.
28. In line 389 you should correct the typo for Na2CO3.
Author reply: Thank you very much for your comments. Corresponding revisions were made in the manuscript.

Round 2
Reviewer 1 Report
I suggest to accept the revised paper.
Author Response
We would like to thank the editor and reviewers for their efforts and suggestions on our manuscript, which have certainly helped to improve the quality of the manuscript.
Reviewer 2 Report
Lines 102-109 should go to methods
In my opinion, the authors should resign (remove) with estimating Fis because I see many problems here. In fact, if Ho=0 then F=1/Ho/He should be 0, but in the paper is not. Also estimating Fis in nonequilibrium populations (high differentiation among populations) always leads to the Wahlund effect and the bias in estimating Fis. Resignation with providing Fis estimates should not decrease the findings of the paper. Also, in this context, I suggest resigning with the estimates of null alleles.
Besides this comment, most other points were well addressed and amended in a new version of the manuscript.
Author Response
First of all, we would like to thank the editor and reviewers for their efforts and suggestions on our manuscript, which have certainly helped to improve the quality of the manuscript. On the basis of the reviewer’s comments, we have revised our manuscript to improve its current quality. Changes / justifications to their comments and suggestions are provided below.
Lines 102-109 should go to methods
Author reply: Thank you very much for your comments. We moved the contents of Lines 102-109 to methods section and removed duplicate contents.
In my opinion, the authors should resign (remove) with estimating Fis because I see many problems here. In fact, if Ho=0 then F=1/Ho/He should be 0, but in the paper is not. Also estimating Fis in nonequilibrium populations (high differentiation among populations) always leads to the Wahlund effect and the bias in estimating Fis. Resignation with providing Fis estimates should not decrease the findings of the paper. Also, in this context, I suggest resigning with the estimates of null alleles.
Author reply: Thank you very much for your comments. We removed Fis and null alleles results and related contents.
